# Heterotrimeric G Protein Signaling in Abiotic Stress

**DOI:** 10.3390/plants11070876

**Published:** 2022-03-25

**Authors:** Yijie Wang, Jose Ramón Botella

**Affiliations:** School of Agriculture and Food Sciences, University of Queensland, Brisbane 4072, Australia; yijie.wang2@uqconnect.edu.au

**Keywords:** heterotrimeric GTP-binding protein, signal transduction, hormonal signaling, abiotic stress tolerance

## Abstract

As sessile organisms, plants exhibit extraordinary plasticity and have evolved sophisticated mechanisms to adapt and mitigate the adverse effects of environmental fluctuations. Heterotrimeric G proteins (G proteins), composed of α, β, and γ subunits, are universal signaling molecules mediating the response to a myriad of internal and external signals. Numerous studies have identified G proteins as essential components of the organismal response to stress, leading to adaptation and ultimately survival in plants and animal systems. In plants, G proteins control multiple signaling pathways regulating the response to drought, salt, cold, and heat stresses. G proteins signal through two functional modules, the Gα subunit and the Gβγ dimer, each of which can start either independent or interdependent signaling pathways. Improving the understanding of the role of G proteins in stress reactions can lead to the development of more resilient crops through traditional breeding or biotechnological methods, ensuring global food security. In this review, we summarize and discuss the current knowledge on the roles of the different G protein subunits in response to abiotic stress and suggest future directions for research.

## 1. Introduction

Environmental cues have a critical impact on plant growth and development [1]. In the face of increasing environmental pollution and climate change, we are experiencing increased incidence of abiotic stresses, such as drought, salinity, heat, cold, heavy metals, ozone, and UV-B radiation affecting crop yields and increasing demands on plant plasticity [2,3,4]. Heterotrimeric guanine-nucleotide-binding proteins (G proteins), composed of α, β, and γ subunits, are a family of highly conserved signaling molecules involved in multiple biological processes [5,6,7,8,9,10]. G proteins transduce signals from membrane receptors synchronizing development and triggering responses to multiple stresses. In animal systems, the Gα subunit is bound to GDP in the inactive state, which is replaced by GTP upon activation by G protein coupled receptors (GPCRs) leading to dissociation of Gα from the Gβγ dimer and activation of independent signaling pathways by each of the two functional subunits.

Plant G proteins have evolved a number of important differences with their animal counterparts [11,12,13]. Plants lack GPCRs and G proteins are associated with receptor-like kinases and regulator of G protein signaling (RGS) proteins [11,14,15]. Aside from the canonical animal looking subunits, plants possess several atypical subunits with important structural and biochemical differences. As an example, extra-large Gα subunits (XLGs) contain a Gα-like C-terminal domain fused to a non-conserved N-terminal region. XLGs are involved in plant defense, root development, and stress responses [16,17,18]. Plants also contain extra-large Gγ subunits with transmembrane and extracellular domains with important roles in crop productivity [19,20,21,22,23]. Finally, some plant Gγ subunits lack the isoprenylation motif present in all known animal Gγs [24]. Most importantly, plant G proteins do not necessarily use the GDP-GTP switch for activation/deactivation of the signaling cycle with both canonical Gα and XLG subunits being able to signal in a GTP-independent manner [25,26]. A mutant *Arabidopsis* canonical Gα subunit, unable to bind guanine nucleotides, successfully complemented several Gα knockout phenotypes, including morphological abnormalities, such as reduced leaf and hypocotyl lengths, and stomatal opening sensitivity to ABA, suggesting that these traits are nucleotide exchange-independent. In contrast the hypersensitivity to ABA during seed germination as well as the reduced stomata density shown by knockout mutants could not be restored to WT levels by the mutant Gα, suggesting that they require GTP binding. The extra-large Gα subunit XLG2 has also proven to function in a GTP-independent manner [25,26]. A vast majority of G protein studies have been performed in the model system *Arabidopsis thaliana*, which contains a canonical Gα subunit (GPA1), three extra-large Gα subunits (XLG1, XLG2 and XLG3), one Gβ subunit (AGB1), two type A Gγ subunits (AGG1 and AGG2), and one atypical type C Gγ subunit (AGG3). Plant G proteins are involved in multiple aspects of plant growth, including seed germination, seedling development, plant morphology, stomatal movement, biotic and abiotic stresses [5,7,27,28,29,30,31,32]. Among their multiple functions, numerous studies have shown that G protein signaling plays an essential role in plant responses to abiotic stresses. This review summarizes the available knowledge on the role of G proteins in plant abiotic stress responses and discusses the possible signaling mechanisms mediated by G protein in abiotic stress.

## 2. Heterotrimeric G Protein Signaling in the Response to Abiotic Stress

Abiotic stresses such as drought, salinity, and extreme temperatures are major environmental factors that affect plant growth and development [1]. In severe cases, abiotic stress can lead to massive crop yield reductions, threatening food security. Upon perception of external stresses, plants reprogram their transcriptome using multiple signal transduction pathways to promote adaptation to environmental conditions and enhance plant survival chances. As important signaling molecules, plant G proteins have been linked to three of the most important abiotic stresses: drought, high salt concentration, and extreme temperatures.

### 2.1. G Protein Signaling in the Response to Drought Stress

Globally, drought is the most significant abiotic stress affecting agricultural production [33]. Terrestrial plants lose water mainly through stomata, therefore precise control of guard cell opening is a critical mechanism to reduce water loss [34]. Even in ideal conditions, plants need to balance CO_2_ acquisition with water loss to maximize photosynthesis. In *Arabidopsis thaliana*, G proteins are involved in a variety of pathways that regulate stomatal closure in response to drought stress, including abscisic acid (ABA), extracellular calmodulin (ExtCaM), and the brassinosteroid-ethylene pathways [35,36,37]. In addition, G protein regulation of leaf developmental processes determines stomatal density, thus affecting the ability of plants to withstand drought conditions [38,39].

#### 2.1.1. G Protein Involvement in the *Arabidopsis* ABA Signaling Pathway

ABA has an essential role in response to drought stress affecting ion fluxes to promote stomata closure and inhibiting stomata opening to enhance plant survival chances [39]. Although the general consensus is that G proteins are involved in *Arabidopsis* ABA signaling and drought response, there is some discrepancy about the roles of the individual subunits.

*Arabidopsis gpa1* (Gα) mutants show a WT response to ABA promotion of stomatal closure but are defective in ABA-mediated inhibition of inward K^+^ channels and pH-independent ABA activation of anion channels. As a result, *gpa1* mutants are hyposensitive to inhibition of stomatal opening by ABA and have higher rates of water loss in excised leaves than WT plants [35,40]. A key component of the ABA response in guard cells is the influx of calcium ions (Ca^2+^) into the cell through activation of Ca^2+^ channels by reactive oxygen species (ROS) produced by the membrane-bound NADPH oxidases *AtrbohD* and *AtrbohF*. Zhang et al. (2011) reported that ABA-mediated activation of Ca^2+^ channel of guard cells in *Arabidopsis gpa1* (Gα) mutants was defective, while ABA-induced ROS synthesis was also disrupted [41]. The authors proposed that GPA1 is required for guard cell ROS production in response to ABA functioning upstream of the NADPH oxidases, *AtrbohD* and *AtrbohF*. Addition of exogenous H_2_O_2_ restored ion channel activation suggesting that GPA1 deficiency inhibits Ca^2+^ channel activation and ROS production due to disruption of ABA signaling (Figure 1). GPA1 has been proven to interact with RD20/CLO3, a member of the calcium-binding protein family caleosin [42]. *RD20/CLO3* transcript levels are strongly induced by drought, salt, and abscisic acid and the *rd20/clo3* exhibit decreased tolerance to drought and salt stresses, prompting the suggestion that RD20/CLO3 acts as a stress-signaling hub controlling multiple plant stress response mechanisms [43,44]. Disruption of the calcium-binding capacity of RD20/CLO3 abolishes the in vivo interaction with GPA1 assays. Comparative analysis of *rd20/clo3*, *gpa1* single mutants, and *rd20/clo3gpa1* double mutant suggested that RD20/CLO3 is a negative regulator of GPA1 [42]. ABA activates the synthesis of sphingosine-1-phosphate (S1P), a signaling sphingolipid involved in the control of guard cell turgor by inhibition of inward K^+^ channels and activation of slow anion channels to promote stomatal closure and inhibit stomatal opening [45,46]. In *gpa1* mutants, ABA-induced inhibition of guard cell inward potassium channels and pH-independent ABA activation of anion channels are disrupted [35,47], and S1P is unable to regulate guard cell ion channels and initiate stomatal closure, indicating that GPA1 acts downstream of S1P to mediate stomatal closure by regulating downstream guard cell ion channel activity, thereby increasing drought tolerance in *Arabidopsis* [47]. G proteins have also been proposed to regulate the activity of the outward-rectifying potassium efflux GORK channels, an essential component of the stress-induced K^+^ loss from the cytosol based on the presence of a conserved consensus protein sequence for G protein binding motif in the GORK protein sequence, although no experimental proof of the interaction has been provided [48]. Phospholipases mediate hormonal signaling in the response to multiple stresses and are involved in ABA signaling [49]. Interestingly, the *Arabidopsis* phospholipase Dα1 (PLDα1) contains a motif with homology to the known Gα-interacting DRY motif found in animal G protein coupled receptors [50]. Interaction studies showed that PLDα1 interacts with GPA1 through the DRY1 motif, and the interaction has important biochemical implications. PLDα1 activity is inhibited by the addition of GPA1 while GTP abolished the inhibitory effect of GPA1 as well as the binding of PLDα1 with GPA1 [50]. PLDα1 mediates ABA effects on stomatal movements through a bifurcating signaling pathway involving a protein phosphatase 2C (PP2C) in one of the branches and GPA1 in the other branch [51]. It was proposed that phosphatidic acid (PA), the product of PLDα1 as well as PLDα1 itself interact with GPA1 to mediate ABA inhibition of stomatal opening. All the above results suggest that GPA1 is a positive regulator of the drought response in *Arabidopsis*.

Like *gpa1* mutants, *Arabidopsis agb1* (Gβ) mutants have been reported to be hyposensitive to ABA inhibition of stomatal opening while displaying wild-type ABA promotion of stomatal closure [40]. Although this result suggests that AGB1 has a positive role in drought stress, overexpression of AGB1 failed to increase ABA sensitivity over WT levels [40]. Further support for a positive role of AGB1 in drought stress comes from studies of the FERONIA pathway which is involved in the regulation of stomatal movement by ABA [52,53]. Immunoprecipitation experiments using anti-AGB1 antibodies and plasma membrane enriched protein extracts identified the receptor-like kinase FERONIA (FER) as an AGB1 interactor [54]. FER ligands include the rapid alkalinization factor (RALF) family of polypeptides and the authors demonstrated that RALF1 inhibits stomatal opening and promotes stomatal closure [54]. Stomatal regulation by the RALF-FER pathway is G protein-dependent and is absent in *agb1* mutants. In addition to AGB1, AGG gamma subunits and XLGs, but not GPA1 are involved in RALF1-mediated stomatal signaling, perhaps using Ca^2+^ as a second messenger [54]. A study by a different group also reported that *agb1* mutants were hypersensitive to drought compared to WT plants [55]. Plant Gβ subunits positively regulate drought tolerance by increasing ROS detoxification. In open contrast with the above results, Xu et al. (2015) reported that *Arabidopsis agb1* mutants have enhanced drought tolerance suggesting that AGB1 might be a negative regulator of the drought response [56]. The same study found that AGB1 can physically interact with the mitogen-activated kinase (MAPK) AtMPK6, a member of several MAPK cascades with multiple roles in plant development, including regulation of mitotic activity in the root apical meristem [56], regulation of shoot branching, hypocotyl gravitropism, and lateral root formation [57], jasmonate signaling [58], and most importantly, regulation of ABA stomatal responses [59]. Upregulation of four ABA-responsive genes, AtMPK6, AtVIP1, AtMYB44, and RD29A, was greatly increased in *agb1* mutants compared to WT plants [56]. In addition, *agb1* mutants showed increased transcription of ABA and proline biosynthesis genes upon drought treatment suggesting that AGB1 inhibits the synthesis of these two essential players in plant drought tolerance [56].

AGB1, does not function in isolation and forms obligate dimers with Gγ subunits [7]. *Arabidopsis* has three Gγ subunits, AGG1, AGG2, and AGG3 with AGG1 and two showing all the hallmarks of animal Gγs and AGG3 containing plant specific features such as a transmembrane and extracellular domains [19,60,61,62]. Similar to the observations in *gpa1* and *agb1* mutants [35,40], *Arabidopsis agg3* mutants stomatal opening and inward K^+^ currents were hyposensitive to ABA, while ABA-mediated promotion of stomatal closure was wild-type [19]. *agg1* and *agg2 mutant*s showed a WT behavior [63], suggesting that AGG3 is the only Gγ subunit participating in the ABA signaling pathway in *Arabidopsis* guard cells. Similarly, *Camelina sativa*, a close relative of *A. thaliana*, overexpressing AGG3 showed hyposensitivity to ABA in seed-related traits but were hypersensitive to ABA in stomatal responses, resulting in increased drought stress tolerance [64].

#### 2.1.2. G Protein Involvement in the *Arabidopsis* ExtCaM Signaling Pathway

Cytosolic calcium [Ca^2+^]_cyt_ is an important second messenger in plants and animals controlling multiple cellular responses [65]. In particular, [Ca^2+^]_cyt_ levels can widely fluctuate in response to hormonal and environmental stimuli in guard cells to promote opening or closing of stomata [66]. Interestingly, a large proportion of the total Ca^2+^ is located outside the cell [Ca^2+^]_ext_ and [Ca^2+^]_ext_ levels can be perceived by sensors such as calcium-sensing receptors in the plasma membrane and extracellular calmodulin to stimulate multiple intracellular signaling pathways [36,67]. Activation of ExtCAM by [Ca^2+^]_ext_ induces increases in [Ca^2+^]_cyt_ and H_2_O_2_ leading to stomatal closure [36]. Experiments using G protein activity modulators suggested that G proteins were involved in the ExtCAM signaling pathway upstream of the increase in [Ca^2+^]_cyt_.

Confirming the pharmacological results, *Arabidopsis gpa1* mutants were defective in ExtCaM-mediated induction of stomatal closure, while transgenic lines overexpressing a constitutively active Gα mutation (cGα) showed an enhanced ExtCaM response [36]. In addition to [Ca^2+^]_cyt_ and H_2_O_2_, ExtCaM-triggered nitric oxide (NO) accumulation, mediated by *AtNOA1*, has a critical role in ExtCaM-induced stomatal closure [68]. In *gpa1* mutants, ExtCaM is unable to induce NO production, whereas NO levels are increased in cGα overexpressing plants, suggesting that ExtCaM-mediated NO accumulation is controlled by GPA1. Furthermore, overexpression of *AtNOA1* rescued the phenotype of *gpa1* mutants suggesting that GPA1 acts upstream of AtNOA1. Overall, the existing data suggests that ExtCaM-signaling involves GPA1-mediated H_2_O_2_ production followed by NO synthesis leading to stomatal closure. In addition, the pea Gβ subunit has also been implicated in the signaling pathway leading to NO-induced stomatal closure during heat and drought stress [69].

#### 2.1.3. G Protein Involvement in the *Arabidopsis* Brassinosteroid-Ethylene Pathway

Aside from ABA, G proteins are involved in the signaling pathways of several other phytohormones [70]. Brassinosteroids (BR) and ethylene promote stomatal closure by induction of H_2_O_2_ and NO levels [37,71]. Ethylene-mediated induction of stomatal closure is dependent on GPA1 signaling as *gpa1* mutants are defective in ethylene-induction of H_2_O_2_ and stomatal closure while transgenic lines expressing either WT or constitutively active cGα show an enhanced response to ethylene [71]. Three of the five known ethylene receptors, ETR1, ERS1, and EIN4 are involved in the guard cell responses to ethylene, while ETR2 and ERS2 do not seem to play an important role [71]. Pharmacological studies with Gα activators revealed a complicated picture with partial or total phenotypic rescue in the *etr1*, *ers1,* and *ein4* mutant backgrounds. CTR1 acts immediately downstream of all ethylene receptors as a negative regulator of the ethylene response. *ctr1* mutants showed elevated H_2_O_2_ levels and constitutive stomatal closure that could be reversed by incubation with Gα inhibitors, suggesting that GPA1 acts downstream from CTR1 [71]. Analysis of mutants for three downstream components of the ethylene response, *ein2*, *ein3,* and *arr2* showed that Gα activators failed to induce stomatal closure thus positioning G proteins upstream of these elements of the ethylene response [71]. BR promotion of stomatal closure seems to be mediated by ethylene [37]. The bioactive 24-epibrassinolide (EBR) induces stomata closure by inducing ethylene synthesis, activation of Gα, and accumulation of H_2_O_2_ and NO. Incubation with EBR enhances the expression of the ethylene biosynthetic genes *AtACS5* and *AtACS9* resulting in ethylene synthesis, which in turn induces H_2_O_2_ synthesis through *AtrbohF* and subsequent NO synthesis finally triggering stomatal closure [37]. Shi et al. (2015) showed that EBR-induction of H_2_O_2_ and NO was defective in *gpa1* mutants but enhanced in transgenic lines overexpressing the constitutively active cGα [37]. Additional mutant and pharmacological data positioned G proteins downstream of ethylene production but upstream of H_2_O_2_ and NO in this signaling pathway, consistent with the results of Ge et al. [71].

Although the BR-ethylene and the ExtCAM pathways seem to share the signaling downstream of GPA1, i.e., induction of H_2_O_2_ and NO synthesis, the signaling upstream of GPA1 and its activation mechanism are still unclear, specially taking into account that GPA1 does not necessarily follow the GTP/GDP activation/deactivation mechanism used in animal systems [25].

#### 2.1.4. G Protein Involvement in the Response to Drought Stress in Food Crops

Many rice G protein subunit genes, including the canonical Gα (RGA1), Gβ (RGB1), and two of the Gγ subunits, RGG1 and RGG2 are induced by drought stress, suggesting a possible role for G proteins in this stress [72,73,74,75]. ABA treatment also induced RGB1 expression while transcript levels for the atypical Gγ subunit DEP1 (also known as qPE9-1) decreased [76]. When transgenic lines containing DEP1 RNA interference constructs were analyzed, silencing of DEP1 resulted in enhanced drought tolerance in rice plants whereas silencing of AGB1 reduced drought tolerance [76]. Consistent with these results, simultaneous overexpression of RGB1 and AGG1 in rice increased the drought tolerance of the transgenic lines by potentiating plant the antioxidant machinery in stress situations [75]. On the other hand, the *rga1*-null mutant, also known as *d1*, showed improved drought resistance compared to WT plants, which was associated with increased stomatal conductance and low leaf temperature in the plants [77,78]. The finding that RGA1 was a negative regulator of mesophyll conductance, with *d1* mutants showing improved photosynthesis, water use efficiency and drought tolerance supported a negative effect for RGA1 on drought. Taken together, the above results suggest that the Gα subunit RGA1 and the Gγ subunit DEP1 negatively regulate ABA signaling and drought adaptation, while the Gβ subunit RGB1 positively regulates the response to drought and ABA. Surprisingly, although the available evidence assigns antagonistic roles for Gα and Gβ in the drought response in rice and *Arabidopsis*, their roles seem to be opposite. The role of the remaining rice G protein subunits in drought resistance remains to be established.

Mulberry (*Morus alba* L.) is a drought-tolerant economic crop, and the roles of several mulberry G protein subunits in drought stress have been studied. Overexpression of *MaGα* in tobacco enhanced drought sensitivity [79]. Extensive analysis revealed that the antioxidant capacity of the transgenic plants was weakened and the content of H_2_O_2_ and O^2-^ was increased. In contrast, overexpression of *MaGβ* and *MaGγ* (γ1 and γ2) increased the expression levels of glutathione peroxidase (POD) and antioxidant genes, and enhanced the drought tolerance of the plants, suggesting that G proteins could regulate drought adaptation by enhancing ROS detoxification [55,79,80]. The available evidence suggests that Gα has a negative regulatory effect on drought stress in mulberry, whereas Gβ and Gγ (γ1 and γ2) are positive regulators.

In contrast to rice and mulberry, the cucumber Gα subunit *CsGPA1* appears to be a positive regulator of the drought stress response since downregulation of *CsGPA1* by RNAi resulted in reduced drought tolerance in the transgenic lines [81]. Transgenic seedlings experienced higher water loss rates in leaves, perhaps as a result of the upregulation of several aquaporin genes, and when subjected to drought stress accumulated higher levels of H_2_O_2_ and malondialdehyde and decreased antioxidant enzyme activities than WT controls. In Chinese white pear (*Pyrus bretschneideri*), Gγ subunit expression show a mixed behavior in response to ABA, with some genes being upregulated while others show reduced expression [82]. In agreement with rice and mulberry, the pea Gβ subunit *PsGβ* is a positive regulator of the drought response with transgenic tobacco lines overexpressing *PsGβ* showing increased tolerance to drought stress [69]. The *PsGβ*-overexpressing lines showed increased NO production during drought stress leading to increased NO-induced stomatal closure. NO production was also enhanced in the transgenic lines in response to ABA and H_2_O_2_ treatment [69].

#### 2.1.5. G Protein Involvement in Developmental Control of Drought Stress

Zhang et al. (2008) reported that *Arabidopsis* Gα is a positive regulator of stomatal development in cotyledons, with *gpa1* mutants showing a reduction in stomatal density and transgenic lines overexpressing a constitutively active GPA1 subunit (GPA1QL) having increased stomatal density [83]. In contrast, *agb1-2* mutants had increased stomatal density while transgenic lines overexpressing *AGB1* contained decreased stomatal density. Stomatal density in *gpa1 agb1* double mutants was the mean of the single *gpa1* and *agb1* mutants, suggesting that GPA1 and AGB1 regulate epidermal stomatal density in an antagonistic manner [83]. The amount of CO_2_ assimilated per unit of water lost by transpiration is known as transpiration efficiency (TE) and achieving an optimal ratio is essential for plant fitness, especially in C3 photosynthesis species [84]. Despite *gpa1* mutants being defective in abscisic acid-induced inhibition of stomatal opening [35,40], they showed increased TE during drought stress and upon ABA treatment [38]. Stomatal density in fully developed *gpa1* mutant leaves was almost half of those observed in WT while stomatal conductance was also reduced. The authors proposed that GPA1 acts as a negative regulator of TE by controlling stomatal conductance and density in leaves [38].

### 2.2. Protein Signaling in the Response to Salt Stress

Salt stress caused by soil salinity is responsible for important agricultural losses worldwide [85]. Plants exposed to salt stress initially experience osmotic stress due to the reduced water potential at the root surface, resulting in impaired water uptake by the plant [86]. In addition, excessive uptake of Na^+^ leads to ionic stress by interfering with the uptake of other ions (such as K^+^) affecting plant growth and even causing cell death [87,88]. The combination of osmotic and ionic stresses can then elicit other secondary stresses including the accumulation of toxic reactive oxygen species (ROS) which cause major damage to cellular structures and essential molecules such as proteins, lipids and nucleic acids [86]. G protein involvement in the salt stress response has been amply documented using mutants and overexpression of G protein subunits in transgenic plants. Expression studies in *Arabidopsis*, rice, pea, and rapeseed have shown upregulation of Gα, Gβ, and Gγ transcript levels under salt stress [72,89,90,91,92,93,94].

Experimental evidence in multiple plant species suggests that the Gα subunit is a negative regulator of salt stress tolerance. *Arabidopsis gpa1* mutants showed enhanced tolerance to salt stress [95], while rice and maize Gα-deficient mutants experienced reduced growth inhibition and senescence in response to NaCl although Na^+^ concentration in Gα mutants was indistinguishable from WT [96]. In rice, a mutagenesis study allowed to isolate a Gα *rga1* mutant (named *sd58*) with enhanced tolerance to salt stress [97]. Proteomic analysis of the mutant showed 332 differentially abundant proteins mostly involved in metabolic pathways, photosynthesis, and reactive oxygen species (ROS) homeostasis [97]. Under salt stress, *sd58* mutants show reduced ROS accumulation, prompting the authors to hypothesize that RGA1 is involved in ROS scavenging during salt stress. Mulberry has a strong capacity to endure abiotic stresses, including drought, salinity, and water logging [80]. Ectopic expression of the mulberry Gα subunit, *MaG**α*, in tobacco decreased the tolerance to salt stress in transgenic lines, providing additional support for a negative role for Gα in salt stress.

In open contrast with the above data, Gα seems to have a positive effect in salt stress tolerance in cucumber and pea. The canonical Gα subunit, *CsGPA1*, in cucumber has been linked to the modulation of aquaporin activity during salt stress by direct interaction with CsTIP1.1, a member of the aquaporin family [98]. Silencing of *CsGPA1* by RNA interference inhibited the expression of *CsTIP1.1* and other aquaporins in roots and leaves under salt stress, reducing cell water content and increasing withering in leaves [98]. Pea contains two canonical Gα isoforms, *PsGα1* and *PsGα2*, and transgenic tobacco plants overexpressing either of them showed increased tolerance to salt stress suggesting that Gα is a positive regulator of salt stress tolerance in pea [94]. Biochemical studies proved that *PsGα1* interacts with phospholipase C (PLCδ) at its C-terminus, in a similar way to the reported interaction between mammalian Gαq and PLCβ, with the interaction increasing the GTPase activity in Gα [94,99]. G proteins have also been involved in the protective role of NO in salt stress [100]. Chemical G protein activators enhanced the accumulation of antioxidant proteins as well as the release of NaCl-induced NO, while G protein repressors had the opposite effect [100].

*Arabidopsis agb1* mutants are more sensitive to salt stress than WT plants, suggesting that AGB1 positively regulates salt stress tolerance in *Arabidopsis* [87,89,101,102]. *agb1* mutants displayed accelerated senescence and impaired development compared with wild type controls when exposed to salt stress conditions [101]. Upon exposure to NaCl, *agb1* mutants accumulated increased levels of Na^+^ than WT as well as decreased amounts of K^+^ suggesting that AGB1 is primarily involved in the ionic toxicity component of salt stress (Figure 1) [87]. Ma et al. (2015) showed that compared to WT plants, *agb1* mutants contained lower chlorophyll levels, reduced proline accumulation and peroxidase activity levels upon exposure to salt stress, while malonaldehyde content and the Na^+^/K^+^ ratio were increased [89]. Transcript levels of genes involved in proline biosynthesis, oxidative stress response, and Na^+^ homeostasis were reduced in *agb1* mutants suggesting that the positive role of AGB1 in stress tolerance was exerted through regulation of multiple genes [89]. Ectopic expression of the rice Gβ subunit (*RGB1*) in transgenic rice lines conferred increased salt stress tolerance, with *RGB1*-overexpressing lines showing reduced electrolyte leakage and higher chlorophyll levels than non-transgenic controls [103]. The expression levels of several stress-related genes were increased in the *RGB1*-overexpressing lines compared to the controls, including enzymes involved in the defense against oxidative stress such as superoxide dismutase (SOD) and ascorbate peroxidase (APX) [103,104]. Coherent with these results, simultaneous overexpression of the *RGB1* and *RGG1* genes in rice resulted in increased salt tolerance and increased antioxidant machinery in the transgenic lines [75]. Overexpression of the mulberry Gβ subunit, *MaGβ*, in tobacco increased salt stress tolerance with overexpressing plants showing higher proline content, and higher activities of the ROS detoxifying enzymes POD, SOD, and catalase (CAT) under salt stress conditions [79]. The two overexpression studies strongly suggest that Gβ subunits enhance salt stress tolerance by improving ROS detoxification. In contrast, overexpression of the pea Gβ subunit in tobacco did not improve salt stress tolerance in the transgenic lines [94].

Several downstream effectors have been linked to Gβ subunits in salt stress. The interplay between AGB1, the receptor-like kinase FER, and its ligand RALF1 in response to salt stress has been investigated and seems to be different from that reported in stomatal movement [54,88]. AGB1 and FER act additively or synergistically under different salt stress conditions, suggesting that the AGB1- and FER-signaling pathways during salt stress overlap but are not in a linear relationship. The AGB1/FER synergism likely occurs through salt-induced ROS production. In addition, RALF1-mediated signaling in the salt stress response seems to be independent of AGB1 [88]. A recent study has shown that the N-MYC DOWNREGULATED-LIKE1 (NDL1) protein physically interacts with AGB1 and acts downstream of AGB1 in salt stress signal transduction [102]. Interestingly, the role of NDL1 in salt stress seems to change depending on the developmental stage of the plant. During germination, NDL1 is a positive regulator of the salt stress response, similar to AGB1, while at the seedling stage this role is reversed [102].

Given that Gβ subunits need to form obligate dimers with Gγs, it is not surprising that G protein γ subunits have been implicated as positive regulators in the salt stress response by stimulating ROS scavenging [72,105]. In rice, the expression of two Gγ subunits, *RGG1* and *RGG2*, was strongly induced by NaCl treatment. Transgenic rice lines overexpressing *RGG1* showed improved salinity tolerance even in the presence of very high salt concentrations without yield penalties in non-stress conditions [105]. Salt stress-induction of transcript levels and enzymatic activities of antioxidant proteins such as APX, CAT, and glutathione reductase (GR) were increased in the transgenic lines compared to non-transgenic controls. In addition, salt-stress-related accumulation of malondialdehyde (MDA), a marker of oxidative stress, and H_2_O_2_ as well as ion leakage were reduced in the transgenic lines while relative water content and proline levels were increased. Yeast two hybrid screening identified a number of RGG1-interacting partners that might be involved in the adaptation to stress conditions. Coherent with the observations in rice, tobacco transgenic lines overexpressing two different Gγ subunits from mulberry also showed increase salt stress tolerance [79].

Overall, the involvement of G proteins in the adaptation to salt stress seems to be mostly exerted by enhancing ROS scavenging through multiple signaling pathways but the exact nature of those pathways is still unclear.

### 2.3. G Protein Signaling in the Response to Temperature Stresses

Temperature stresses comprise heat and cold stress, both of which can affect normal crop development and result in substantial yield reduction. Expression of multiple G protein subunits is altered by temperature stresses although it follows species-specific patterns. In pea, the expression levels of genes encoding (*PsGα1* and *PsGα2*) and Gβ (*PsGβ*) were significantly induced by heat treatment [94], with *PsGβ* being also induced by low temperature [69], while in Chinese pear (*Pyrus pyrifolia*) six out of eight Gα genes were upregulated by heat stress [106]. In contrast, rapeseed (*Brassica napus*) genes encoding for Gα (*BnGA1*), Gβ (*BnGB1*) and Gγ (*BnGG2*) subunits were down-regulated by heat and cold stress [91,92,93]. Rice has an even more complicated expression pattern with RGA1, RGB1, RGG1, and RGG2 up-regulated by cold stress while heat stress results in the downregulation of RGA1, upregulation of RGG1 and RGG2, and has no effect on RGB1 expression [72,73,74,90].

In *Arabidopsis gpa1* mutants, the expression of 144 abiotic stress-related genes, including heat and cold stress, was significantly altered, suggesting that GPA1 may be involved in the response to temperature stress [95,107]. Further analysis of *gpa1* mutants showed increased tolerance to cold and, in to lesser extent, heat stress suggesting a negative role for GPA1 in the adaptation to cold and hot temperatures [95]. In contrast with the *Arabidopsis* results, transgenic tobacco plants overexpressing a pea Gα subunit showed increased tolerance to heat stress suggesting a positive role for the pea Gα subunit in the adaptation to heat [94]. Transgenic tomato lines either overexpressing the *LeGPA1* Gα subunit or containing RNA interference constructs have been characterized for cold tolerance [108]. *LeGPA1*-overexpressing lines showed increased tolerance to cold stress than WT controls while *LeGPA1*-silenced lines exhibited decreased cold tolerance. Biochemical characterization of the plants showed that under cold stress conditions, *LeGPA1* overexpressing plants accumulated less H_2_O_2_ and superoxide ions (O^2−^) than WT perhaps due to the higher activities observed for the antioxidant enzymes SOD, POD, and CAT, whereas the proline and soluble sugar content in the transgenic lines were higher than WT. In contrast, *LeGPA1*-silenced plants showed reduced SOD, POD, and CAT activities with a concomitant increase in ROS upon cold stress compared to WT controls. The inducer of CBF expression (ICE)-C repeat binding factor (CBF)-cold-responsive (COR) (ICE-CBF-COR) pathway controls signaling upon cold perception to regulate downstream genes involved in the production of osmoregulatory compounds [109]. Cold-treated *LeGPA1*-overexpressing lines showed enhanced levels of *LeICE1*, *LeCBF1,* and *LeCOR413PM2* gene expression over WT plants resulting in elevated levels of proline and soluble sugar to protect against cellular damage, suggesting that *LeGPA1* has a positive role in the adaptation to cold stress in tomato [108].

Microarray analysis of the rice Gα-deficient mutant Daikoku 1 (*d1*) identified 2270 differentially expressed genes (DEGs) with a large subset of them (1498) involved in metabolic and developmental processes related to drought, salt, heat, and cold stresses [90]. A total of 1690 and 1688 DEGs were related to heat stress and cold stress respectively. In silico analysis of the *RGA1* promoter revealed the presence of several stress-related cis-regulatory elements [73]. Although very little is known about the signaling cascades regulated by G proteins in temperature stress, an important insight has come from the discovery and characterization of COLD1, a critical factor in the adaptation to cold temperatures in rice [110]. Mutations in COLD1 enhance sensitivity to cold while transgenic lines overexpressing COLD1 show significantly improved chilling tolerance. Map-based cloning identified COLD1 as a regulator of G protein signaling co-located in the plasma membrane and the endoplasmic reticulum. COLD1 acts as a cold temperature sensor, perhaps by cold-induced structural changes in the protein, stimulating intrinsic GTPase activity in RGA1 and the subsequent activation of Ca^2+^ channels initiating signaling cascades to mediate the plant response to cold stress [110,111].

Transgenic rice lines overexpressing the Gβ subunit *RGB1* showed enhanced tolerance to heat stress as well as a combination of simultaneous heat and salt stress compared with control lines, inferring a positive effect of RGB1 in heat adaptation [103]. Heat-induction of genes encoding heat shock proteins such as OsHSP1 and OsHSP2, which are involved in protection against heat stress [112], was more prominent in the transgenic *RGB1* overexpressing plants compared to WT controls [103]. In a similar way, the enhanced induction of several genes for ROS scavenging enzymes observed in transgenic lines could contribute to the increased heat tolerance. No data were provided for cold tolerance in the transgenic *RGB1* overexpressing lines, but they showed enhanced induction of *COR47*, a known provider of cold stress protection [113]. It is important to remark that these studies were performed in plants at the seedling stage and no data were provided for mature plants in the reproductive stage. Consistent with the results in rice, overexpression of a pea Gβ subunit in transgenic tobacco increased heat tolerance [94].

Gγ subunits are divided into A, B, and C types based on their protein structural characteristics. Type A Gγ subunits are very similar to their animal counterparts, while type B lacks the obligatory isoprenylation motif found in all animal Gγs [24,114]. Type C Gγ subunits are quite atypical and unique to plants, containing a transmembrane domain and an additional C-terminal region located in the extracellular space [19,24,62]. The expression of rice type A (*RGG1*) and type B (*RGG2*) Gγ subunits is strongly induced by heat and cold stresses while the expression of the cucumber type C (*CsGG3.2*) Gγ subunit is strongly induced by cold [68]. Rice lines overexpressing *RGG1* showed enhanced salt stress tolerance but unfortunately no temperature stress experiments were performed on the transgenic lines [105]. In cucumber, overexpression of *CsGG3.2* provided tolerance to cold stress in the transgenic lines [115]. The C repeat binding factor (CBF) regulates the induction of cold-responsive genes during chilling adaptation to induce biochemical and physiological adaptation responses [116,117]. *CsGG3.2* overexpressing plants showed increased expression of CBF family genes as well as CBF-dependent downstream genes resulting in higher enzymatic activities for antioxidant proteins such as SOD, CAT, POD, and GR. In cold stress conditions, transgenic cucumber plants produce less oxygen free radicals, H_2_O_2_ and MDA than WT controls suggesting that overexpression of *CsGG3.2* provides strong protection to the oxidative damage caused by cold stress. A recent report described the map-based cloning of a natural quantitative trait locus, TT2, conferring thermotolerance in rice [118]. Loss of TT2 function abolishes the heat-triggered reduction in leaf cuticular wax, resulting in enhanced wax retention and improved thermotolerance. Interestingly, TT2 is a type C Gγ subunit, commonly known as GS3, that plays a major role in grain size [21,23]. TT2/GS3 mediates heat-induced Ca^2+^ influx into the cytosol and the reduction of Ca^2+^ in the mutants disrupts the CaM-mediated regulation of transcription factors that trigger the reduction in leaf cuticular wax [118].

### 2.4. G Protein Signaling in the Response to Other Stresses

In addition to the above-discussed abiotic stresses, heterotrimeric G proteins are involved in ozone (O_3_), UV-B, and heavy metal stresses. 

Ozone is a major air pollutant and has deleterious consequences for agricultural crops [119]. *Arabidopsis gpa1* mutants are more resistant to O_3_ damage than WT plants while *agb1* mutants are hypersensitive [120]. Exposure to O_3_ induces a biphasic oxidative burst. The fast first phase is aimed for intercellular signaling, with ROS initially produced in the chloroplasts of guard cells. In the second phase, ROS production expands to adjacent cells triggered by extracellular ROS signals produced by *AtrbohD* and *AtrbohF* resulting in cell death. Biochemical dissection of the process showed that the early component of the oxidative burst, arising primarily from chloroplasts, requires signaling through the G protein heterotrimer while the activation of membrane-bound NADPH oxidases necessary for intercellular signaling and cell death is mediated by GPA1 [120]. In *Arabidopsis*, exposure to UV-B induces H_2_O_2_ synthesis and activates the enzyme nitrate reductase inducing the production of NO and leading to stomatal closure. This process requires GPA1-mediated signaling to *AtrbohD* and *AtrbohF*, thereby activating downstream effectors and enhancing the effect of UV-B [121]. Two different groups have reported a connection between type C Gγ subunits and tolerance to heavy metals [122,123]. *Camelina sativa* transgenic plants overexpressing the *Arabidopsis* Gγ *AGG3* showed enhanced tolerance to high cadmium levels. Screening of yeast cells transformed with a rice cDNA expression library for enhanced Cd tolerance identified the rice type C Gγ subunit DEP1 [122]. The C-terminal region of type C Gγs is extremely rich in cysteine residues although the functional significance of such a high Cys content is unknown [20,23]. Expression of the entire DEP1 protein and the C-terminal (Cys-rich) region in yeast enhanced Cd tolerance while expression of the N-terminal region had no effect, suggesting that the tolerance is linked to the C terminus. Transgenic *Arabidopsis* plants overexpressing either the complete rice DEP1 or its C-terminal region also showed enhanced Cd tolerance but the expression of the DEP1 N-terminus showed WT levels of tolerance [122]. The authors proposed that the observed enhanced tolerance was due to the Cys-rich region acting as a trap for Cd ions but the fact that *gpa1, agg1* and type C Gγ *agg3 Arabidopsis* mutants are hypersensitive to Cd argues against this hypothesis and suggests a role for G protein-mediated signaling. Rice RGA1 expression is induced by several heavy metals including cadmium, zinc, manganese, arsenite, arsenate, and lead but *RGB1* expression levels do not show significant changes [73,74].

## 3. Discussion

Heterotrimeric G proteins mediate a wide range of developmental processes in plants as well as the responses to the most important abiotic stresses. Due to their inherent functional nature, with two independent functional subunits (Gα and the Gβγ dimer), G proteins can simultaneously mediate multiple signaling pathways, and there may be overlaps and interactions between different pathways. Some upstream elements as well as some downstream components of the signaling pathways have been established, but the overall picture is still quite incomplete. Future work is needed to establish all the components of the G protein-mediated stress signaling pathways. In addition, there are some discrepancies about the positive or negative roles of the subunits in different plant species. Although certainly interesting, overexpression studies need to be cautiously interpreted for functional inferences in the case of G proteins, given their mode of action. For example, overexpression of the Gγ subunit *RGG1* in rice enhanced salinity tolerance [105]. While a logical inference is that RGG1 is a positive regulator of the salinity response, rice has five different Gγ subunits with specific functions. A vast excess of one of them, like RGG1, will increase the amount of RGB1/RGG1 dimers in the cell but will also vastly reduce the amount of other possible Gβγ dimer combinations thus reducing the signaling mediated by those combinations. This fact has been experimentally proven in *Arabidopsis* as overexpression of a truncated, defective, Gγ subunit produced phenotypes similar to α and β subunit knockouts [124]. The same logic applies to Gα subunits as overexpression of a specific Gα will sequester most available Gβγ dimers and reduce the amount of heterotrimers containing the remaining Gα subunits.

A common pattern arising from multiple studies is that G proteins can enhance adaptation to abiotic stresses by reducing cellular damage by reactive oxygen species, either by limiting ROS production or by enhancing the detoxification mechanisms. Although there is ample evidence of the involvement of G proteins in abiotic stresses, most of the knowledge has been produced in a few species and more studies in economically important crops are urgently needed. G proteins have strong biotechnological potential and some very promising results have shown that altering G protein content can confer improved stress tolerance. As an example, *RGG1* overexpression strongly improved salinity tolerance in rice but did not convey yield penalties in non-stress conditions [105].

## Figures and Tables

**Figure 1 plants-11-00876-f001:**
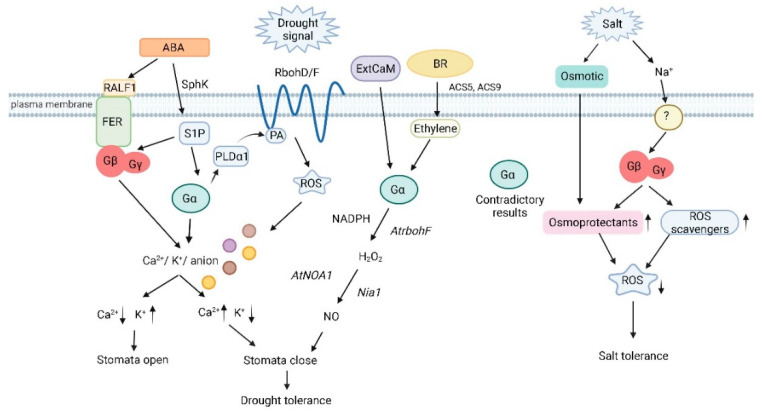
Heterotrimeric G protein signaling in plant drought and salinity stress (Created with BioRender.com on 17/03/2022). Gα deficiency inhibits Ca2^+^ channel activation and ROS production due to disruption of ABA signaling. Gα acts downstream of S1P to regulate stomatal closure by modulating inward K^+^ channels and slow anion channels. PA and PLDα1 interact with Gα to mediate ABA inhibition of stomatal opening. Binding of PA to Rboh at the N-terminal cytoplasmic region results in the production of ROS. Stomatal regulation by the RALF-FER pathway is associated with Gβ. Gα participates in ExtCaM-induced stomatal closure of guard cells by regulating NO synthesis, and NO synthesis is dependent on H_2_O_2_ produced by NADPH oxidases through the action of the nitrate reductase Nia1. BR induces the expression of ACS5 and ACS9 to initiate ethylene synthesis, which signals through Gα to synthesize H_2_O_2_ and before the nitric synthase NOA1 induces the production of NO. Several groups have proven the involvement of Gα in salt stress although the results are contradictory with positive as well as negative roles being proposed for this subunit. Gβ regulates the combination of osmotic and ionic stresses during salt stress by increasing levels of proteins involved in ROS detoxification and osmoprotectant compounds. SphK: sphingosine kinase; RALF1: rapid alkalinization factor 1; FER: receptor-like kinase FERONIA; S1P: sphingosine-1-phosphate; PLDα1: phospholipase D α1; PA: phosphatidic acid; RbohD/F: NADP oxidases RbohD and RbohF; ROS: reactive oxygen species; ExtCaM: extracellular calmodulin; AtNOA1: nitric oxide; Nia1: nitrate reductase.

## Data Availability

Not applicable.

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
