# Peer review of "Heterotrimeric G Protein Signaling in Abiotic Stress"

_plants, 2022, doi:10.3390/plants11070876_

Round 1

Reviewer 1 Report

The manuscript entitled “Heterotrimeric G Protein Signaling in Abiotic Stress” is a well written review summarizing the role of plant G proteins in responses to various types of abiotic stresses: drought, salinity, heat, cold, heavy metals, and others.

Dear All,

Please, find my comments and suggestions.

General suggestions:

  1. Write gene and transcript names in italics, while proteinnames using normal font throughout the text.
  2. Write Latin species names in italics, whether they are written in full or abbreviated version.
  3. There are some recent publications in the topic, which are interesting. I suggest to add them.

  1. Introduction:

Instead: “with receptor kinases” write “with receptor-like kinases”

I suggest to write a sentence or two about the names of Arabidopsis G protein subunits in the introduction part. It would be easier to read further parts of the article.

I suggest to write shortly about possible GTP-independent mechanisms of G protein action. Moreover, about the extent of GTP-dependent functioning.

Next, it would good to mention whether GTP levels change under stress conditions, and whether it could potentially affect G proteins action? Could that be a driving point for the evolution of plant G proteins towards GTP-independence (at least when compared to animals)? This part could be mentioned in the Discussion.

  1. Heterotrimeric G protein signaling in the response to abiotic stress

 Instead:As important signaling molecules, plant G proteins have been linked to three of the most important abiotic stresses: drought, salt and temperature.”, e.g.: “As important signaling molecules, plant G proteins have been linked to three of the most important abiotic stresses: drought, high salt concentration and extreme temperatures.”

2.1. G protein signaling in the response to drought stress

I suggest to use other words than “maintain fitness”

“In addition, G protein regulation of leaf developmental processes determines stomatal density, thus affecting the ability of plants to withstand drought conditions [17,39,40].” The citation 17 is not really relevant here.

2.1.1. G Protein involvement in the Arabidopsis ABA signaling pathway

Once you write:Arabidopsis gpa1 () mutants show a WT response to ABA promotion of stomatal closure but are defective in ABA-mediated inhibition of inward K+ channels”, and later: “In gpa1 mutants, ABA-induced activation of potassium and anion channels is disrupted”. These two sentences seem to be contradictory. 

Figure 1

I suggest to increase the quality of the picture. The scheme is nice but after printing, the quality is quite low.

Instead: “Stomatal closure signaling pathways involving heterotrimeric G proteins in Arabidopsis.”, write “Stomatal closure signaling pathways involving heterotrimeric G proteins in Arabidopsis in response to drought stress”. If you do not want to write in such way then in the figure, instead of “drought signal” write “abiotic signal” or even just “signal”. However, it is inappropriate to make such generalization.

I suggest to improve the description of the figure. Please, keep the same naming for G protein subunit in the text describing the figure and in the figure. If you decided to write GPA1 (or agb1), give the same name for the subunit in the figure, or at least write in the text in brackets the other name.

You write: “GPA1 deficiency inhibits Ca2+ channel activation and ROS production due to disruption of ABA signaling” but on the scheme ROS production is not shown to be dependent on GPA1. It has to be improved.

“ABA-responsive genes are significantly up-regulated in the agb1-2 mutant after ABA treatment or drought stress (AtMPK6, AtVIP1 and AtMYB44).” It does not correspond well with the scheme as AGB1 is shown to promote the expression of these genes. It would also be interesting if you propose here an explanation for such result.

It is not clear what represent the “circles” in the middle of the scheme.

Write explanation for the abbreviations used, e.g.: SphK, etc.

There are some parts of the scheme that are not described in the text. I suggest to either remove them from the scheme or to describe them.

This figure is very important to understand the text. Thus, I suggest to describe the elements that are presented on the scheme not only in the text but in more detail in the description of the figure.

2.1.2 G protein involvement in the Arabidopsis ExtCaM signaling pathway

“In gpa1 mutants, ExtCaM is unable to induce NO production, whereas NO levels are increased in cGα overexpressing plants”. What mean here cGα – constitutive overexpression? Try to write first full name and then the abbreviated in bracket.

2.1.4 G protein involvement in the response to drought stress in food crops

“subunit DEP1 (aka qPE9-1)” – I guess, you mean just “subunit DEP1 (qPE9-1)”

“On the other hand, the rga1-null mutant d1 showed improved drought resistance compared to WT plants, which was associated with stomatal conductance and low leaf temperature in the plants” – please, write in brackets what means mutant d1, and write with what kind of change in stomatal conductance there was the association.

“The finding that RGA1 was inversely correlated” – what about the RGA1 was inversely correlating with mesophyll conductance? I suggest to rewrite this sentence.

“When transgenic lines containing DEP1 RNA interference constructs were analyzed, silencing of DEP1 resulted in enhanced drought tolerance in rice plants whereas silencing of AGB1 reduced drought tolerance [71].” – instead of AGB1 --> RGB1

Overexpression of MaGα” – in my opinion Ma should be written in italics (and always Latin name [abbreviated or full name]). Since we talk about overexpression the name Gα should also be written in italics. I recommend to apply it throughout the text.

2.2. protein signaling in the response to salt stress – please, add “G” in the title

Instead: “In rice, a mutagenesis study isolated a Gα rga1 mutant (sd58) with enhanced tolerance to salt stress”, e.g.: “In rice, a mutagenesis study allowed to isolate a Gα rga1 mutant (sd58) with enhanced tolerance to salt stress”.

From “Transgenic rice lines overexpressing RGG1 showed improved salinity tolerance in even in the presence of very high salt concentrations without yield penalties in non-stress conditions [96].”, please remove the “in” before “even”.

2.3. G protein signaling in the response to temperature stresses                 

while in Brassica napus the opposite behavior was reported with Gα, Gβ and Gγ subunits, BnGA1, BnGB1 and BnGG2, down-regulated by heat and cold stress [83–85].” – please, write Brassica napus in italics, and modify the further part of this sentence as it is grammatically incorrect.

“Rice has an even more complicated expression pattern with RGA1, RGB1, RGG1 and RGG2 up-regulated by cold stress while heat stress results in downregulation of RGA1, upregulation of RGG1 and RGG2 and has no effect on RGB1 expression [68–70,82,97].” – please, be careful with the citation 97 – it is for cucumber

“Finally, the tomato Gα LeGPA1 is upregulated by both heat and cold stresses [98].” – I believe in this particular article the results are only for room temperature and four degrees treatments. I suggest to remove that part since you describe the results from this publication later.

“Heat-induction of genes encoding heat shock proteins such as OsHSP1 and OsHSP2, which are involved in protection against heat stress, was more prominent in the transgenic RGB1 overexpressing plants compared to WT controls [103].” – I believe these results are not presented in the cited publication.

“The expression of rice type A (RGG1) and type B (RGG2) Gγ subunits is strongly induced by heat and cold stresses while the expression of the cucumber type C (CsGG3.2) Gγ subunit is strongly induced by cold [68,106].”  – I guess the citation 97 is the one that you wanted to introduce, instead of 106.

Please, keep in mind that you describe the data from publication 97 later in the text. Maybe you could remove it from here.

2.4. G protein signaling in the response to other stresses

„AtrbohD and AtrbohF” – please, write AtRbohD and AtRbohF when writing about proteins

“Biochemical dissection of the process showed that the early component of the oxidative burst, arising primarily from chloroplasts, requires signaling through the G protein heterotrimer while the activation of membrane-bound NADPH oxidases necessary for intercellular signaling and cell death is mediated by GPA1 [110].” – it is the first time when you write about these NADPH oxidases, and you place their names in the picture where the signal (for stomata closure) is drought. That might be slightly confusing why you write about them now and placed their names when describing G signaling upon different signal. I suggest to write about the results for AtRbohD and AtRbohF – mediated ROS production in response to drought earlier, and possibly, refer to these results here, trying to pinpoint differences/similarities to ozone induced, AtRbohD and AtRbohF – mediated ROS production. Furthermore, is stomata closure a part of the AtRbohD and AtRbohF – mediated ROS production in response to ozone?

“G protein signaling has also been reported as being upstream of NO biogenesis in Maize [106].” – that should go, I believe, to the part where you write about G proteins in signaling in response to salt stress. I would suggest then to refer to the similar results obtained in the case of drought stress.

“Two different groups have reported a connection between type c Gγ subunits…” – please, write type C (with capital letter).

References

  1. There is some visible problem with “et al.” 27 and 32. These are the same citations

Reviewer 2 Report

The presented manuscript describes the influence of plant G proteins on signaling and plant response to selected abiotic stresses, in particular drought, salinity and temperature. The discussed problems are well described in terms of logic and content. They constitute a valuable work describing the participation of G proteins in signaling pathways. However, I have a few critical comments, the introduction of which will definitely increase the value of the presented manuscript. 

A large part of the work is a description of the effects observed in mutants. I believe that the compilation of this information in a table will provide a good summary of what was presented in the manuscript. Furthermore, I think it would be beneficial to introduce a Figure 2 (similar to Figure 1) which would show the contribution of G protein to salinity response. This would make it easier to understand the fragment of text where the authors compare those two responses. I also believe that the work would become more relevant if the share of literature from the last 5 years was greater. Currently, little more than 18% of the used references have been published in the last 5 years, and almost 40% of the literature is older than 10 years. I think that review publications in particular should focus on the latest scientific findings.  

I also have some editorial notes: 

on page 2 - Arabidpsis thaliana should be written in italics; on page 9 - Maize should be written with a lowercase letter (it is middle of the sentence); 

for some plants, the authors use only common names, while others are also referred to by a Latin name (e.g. tomato, mazie). The nomenclature should be unified, especially in the subsection 2.2 

In the subsection 2.2, the authors use the complete words "alpha, beta, gamma" instead of symbols; this is the only place in the manuscript where they are written like this. 

The description of Figure 1 should provide an explanation of the abbreviations used in the diagram. 
